# Effectiveness, Adoption Determinants, and Implementation Challenges of ICT-Based Cognitive Support for Older Adults with MCI and Dementia: A PRISMA-Compliant Systematic Review and Meta-Analysis (2015–2025)

**DOI:** 10.3390/healthcare13121421

**Published:** 2025-06-13

**Authors:** Ashrafe Alam, Md Golam Rabbani, Victor R. Prybutok

**Affiliations:** 1Department of Information Science, University of North Texas, Denton, TX 76207, USA; mdrabbani@my.unt.edu; 2Department of Information Technology and Decision Sciences, G. Brint Ryan College of Business, University of North Texas, Denton, TX 76201, USA; victor.prybutok@unt.edu

**Keywords:** dementia, mild cognitive impairment, digital health, cognitive support, technology adoption, digital literacy, ICT, healthcare technology, assistive technology

## Abstract

**Background:** The increasing prevalence of dementia and mild cognitive impairment (MCI) among the elderly population is a global health issue. Information and Communication Technology (ICT)-based interventions hold promises for maintaining cognition, but their viability is affected by several challenges. **Objectives**: This study aimed to significantly assess the effectiveness of ICT-based cognitive and memory aid technology for individuals with MCI or dementia, identify adoption drivers, and develop an implementation model to inform practice. **Methods**: A PRISMA-based systematic literature review, with the protocol registered in PROSPERO (CRD420251051515), was conducted using seven electronic databases published between January 2015 and January 2025 following the PECOS framework. Random effects models were used for meta-analysis, and risk of bias was assessed using the Joanna Briggs Institute (JBI) Critical Appraisal Checklists. **Results**: A total of ten forms of ICT interventions that had proved effective to support older adults with MCI and dementia. Barriers to adoption included digital literacy differences, usability issues, privacy concerns, and the lack of caregiver support. Facilitators were individualized design, caregiver involvement, and culturally appropriate implementation. ICT-based interventions showed moderate improvements in cognitive outcomes (pooled Cohen’s *d* = 0.49, 95% CI: 0.14–1.03). A sensitivity analysis excluding high-risk studies yielded a comparable effect size (Cohen’s *d* = 0.50), indicating robust findings. However, trim-and-fill analysis suggested a slightly reduced corrected effect (Cohen’s *d* = 0.39, 95% CI: 0.28–0.49), reflecting potential small-study bias. Heterogeneity was moderate (I^2^ = 46%) and increased to 55% after excluding high-risk studies. Subgroup analysis showed that tablet-based interventions tended to produce higher effect sizes. **Conclusions**: ICT-based interventions considerably enhance cognition status, autonomy, and social interaction in older adults with MCI and dementia. To ensure long-term scalability, future initiatives must prioritize user-centered design, caregiver education, equitable access to technology, accessible infrastructure and supportive policy frameworks.

## 1. Introduction

A global increase in the aging population has accounted for the increased prevalence of age-associated cognitive impairments like mild cognitive impairment and dementia, with over 55 million individuals in the world recently estimated to have dementia [1,2]. As mental and functional abilities decline, many older adults express a strong preference to age in place and remain in their homes [3]. However, cognitive and functional impairment present enormous challenges to independence in daily living [4]. This new public health challenge has sparked interest in large-scale, non-pharmacological interventions to facilitate care and improve quality of life. Among these approaches, Information and Communication Technology (ICT) has been viewed as a promising solution. In response to this emerging public health concern, attention has increasingly turned toward large-scale, non-pharmacological strategies aimed at enhancing care and improving quality of life. Among these, ICTs have enhanced the quality of life of older adults with dementia. ICTs range from digital devices, such as smartphones, tablets, and wearables, to telemedicine equipment, intending to facilitate communication, stimulate cognition, and assist activities of daily living (ADLs) among older adults [5,6,7]. These technologies encompass a wide array of digital devices, including smartphones, tablets, wearables, and telemedicine systems, designed to stimulate cognition and bridge communication gaps.

While awareness of gerontechnology has increased since 2014 [8], particularly in high-income nations, its actual adoption among people with dementia has been hindered by barriers such as low digital literacy, usability challenges, and inadequate caregiving support [9,10]. These factors hinder the practical implementation of digital interventions in real-world settings.

The COVID-19 pandemic significantly accelerated the adoption of digital health interventions, transforming the way older adults and their caregivers interact with technology. During that period, digital platforms enabled the continuation of remote clinical assessments, mHealth communication, virtual therapy, and caregiver-mediated support [11,12,13]. However, recent evidence from 2022 to 2023 has revealed persistent inequalities in access to digital tools, highlighting the need for inclusive digital health strategies for aging populations [14,15,16].

Even as ICT use becomes more widespread during and after the pandemic, data indicate that the actual adoption of cognitive-specific technologies remains suboptimal. For instance, although more than half of older adults with dementia or MCI do utilize smartphones or tablets, a small minority (9.76%) utilize programs or apps aimed explicitly at memory or cognitive assistance [17,18]. Barriers include cognitive limitations, a lack of familiarity with ICT systems, and inadequate support from caregivers or health professionals [19,20,21]. On the other hand, individuals with early-stage cognitive decline (MCI) have shown interest in self-management tools and ICT-supported social interaction [22,23].

In this context, caregivers serve as essential facilitators in helping older people with dementia integrate such tools into their daily routines and overcome usage barriers [24,25,26]. Recent studies merge conventional assistive technologies with those designed to support cognitive processes, such as attention, memory, and executive function. Although digital assistive technologies (DATs), such as GPS trackers and fall detectors, support mobility and ensure data security and safety [27], the use and effectiveness of ICT-based cognitive support interventions remain less explored, particularly in everyday life and community-based settings [28].

Moreover, usability problems, cultural insensitivity, and the lack of incorporation of caregiver experience in studies are among the other barriers to implementation [29,30,31]. Addressing these challenges requires a nuanced understanding of the technologies involved, the support systems surrounding their use, and the lived experiences of users and caregivers. Several studies have investigated how caregivers’ perceptions, digital literacy, and the sociotechnical context, such as Facebook and Twitter, significantly influence the adoption of technology among older adults [32]. At the same time, some caregivers and healthcare professionals remain skeptical, assuming older adults with dementia or MCI cannot effectively use such tools, which further hinders their adoption [7,33]. These misconceptions emphasize the importance of tailoring ICT solutions to users’ needs, capabilities, and preferences.

This review aims to systematically examine ICT-based cognitive support interventions that differ from general care or safety technologies in enhancing memory, attention, problem-solving, and day-to-day functioning among older adults with mild cognitive impairment (MCI) or dementia. The interventions evaluated in this review include mobile memory aids, tablet-based cognitive training, telehealth-delivered cognitive behavioral therapy (CBT), wearable cognitive monitoring devices, and AI-based conversational agents. By focusing exclusively on cognitive support functions, this study addresses a significant gap in the literature and contributes differentiated, post-pandemic insights into the evolving field of dementia care technology.

Since previous review studies have broadly assessed digital health or assistive technologies, this study reviews ICT interventions that support cognitive functioning in older adults with dementia or mild cognitive impairment (MCI). Covering literature from 2015 to 2025, it incorporates the most recent trends in telehealth, mHealth, and caregiver-facilitated digital care. The study also aims to explore usability, adoption barriers, and caregiver roles, providing a framework to design scalable, inclusive, and demand-driven, user-centered ICT solutions to support cognitive aging.

The research questions are as follows:What types of ICT technologies are most used in cognitive support interventions?How do ICT-based cognitive and memory support tools affect the mental health, daily activities, and independence of older adults?What are the long-term sustainability and implementation challenges of these interventions in real-world settings?What are the key enablers and barriers that influence the usability and adoption of technology-based cognitive support tools among older adults?

The systematic review and meta-analysis aim to address the above questions and provide a comprehensive understanding of how ICT can facilitate the management of dementia-related decline. The aim will be to direct the development of evidence-based practice that enhances the lives of older people with dementia and MCI.

## 2. Materials and Methods

The systematic literature review followed the PRISMA guidelines [34]. The PECOS framework defines inclusion criteria [35]. The eligibility criteria for systematic reviews and meta-analyses are based on the use of the Population, Exposure, Comparator, Outcome, and Study design (PECOS) framework to ensure methodological rigor and relevance, as follows.

Inclusion and exclusion criteria were defined based on the PECOS framework. Studies were eligible if they examined the use of ICT-based interventions targeting cognitive or psychosocial outcomes in older adults (aged ≥55) diagnosed with mild cognitive impairment (MCI), dementia, or Alzheimer’s disease. Acceptable study designs included randomized controlled trials (RCTs), cohort studies, cross-sectional analyses, mixed-methods studies, and systematic reviews published between 1 January 2015, and 31 January 2025. Eligible exposures included digital cognitive training tools, telehealth platforms, AI-based diagnostic systems, assistive technologies (e.g., GPS tracking, wearables), mHealth applications, and serious games. Comparators involved standard care, paper-based cognitive training, or no intervention. The primary outcomes of interest included changes in cognitive performance (e.g., memory recall, executive function), behavioral and psychological symptoms of dementia (e.g., depression, anxiety), activities of daily living (ADLs) and instrumental activities of daily living (IADLs), and social engagement through technology-mediated interaction. Secondary outcomes involved usability, acceptability, and adherence to ICT interventions.

Studies were excluded if they involved participants aged <55 years, focused on non-ICT interventions (e.g., pharmacological treatment, diet modification, conventional caregiving), or did not evaluate cognitive, functional, or psychosocial outcomes. Editorials, preprints, conference abstracts, and studies published before 2015 were also excluded due to limited data quality or because they were of historical relevance only.

### 2.1. Summary of Inclusion and Exclusion Criteria

A summary of the detailed inclusion and exclusion criteria, organized by the PECOS framework, is summarized in Table 1:

### 2.2. Study Selection Process

The included studies were conducted over the last 10 years (1 January 2015 to 31 January 2025) using the electronic databases PubMed, Web of Science, PsycINFO, Scopus, ProQuest, IEEE, and EBSCOhost. Establishing the study aim, systematically retrieving and assessing pertinent material, and synthesizing and analyzing data are the three primary steps that comprise the systematic literature review method. A standardized search procedure was employed across the databases to identify, review, and evaluate studies on the implementation of ICT in dementia care management. The study questions corresponded with the data acquisition and analysis. The PRISMA checklist guarantees consistency and reliability in obtaining an in-depth analysis. The whole selection process of the study papers in this systematic review is shown in Figure 1 as follows:

The search term (Table 2) was created using the keywords and the Boolean operators “AND” and “OR”:

### 2.3. Data Extraction

Data was retrieved and extracted from multiple databases, including PsycINFO, PubMed, Scopus, Web of Science, ProQuest, IEEE, and EBSCOhost. Firstly, the records were imported into Zotero, and duplicate articles were removed. Two authors worked independently to screen the titles and abstracts, identifying studies based on the established inclusion and exclusion criteria. The third reviewer was responsible for resolving any discrepancies that arose. Thus, the final selections were confirmed.

### 2.4. Selected Study

The systematic literature review initially generated 3350 records from specified databases. Then, the removal of 1820 duplicate records and 1730 unique records was screened based on titles and abstracts, resulting in the exclusion of 1655 studies, which were deemed to be irrelevant or failed to meet the inclusion criteria. Full texts were retrieved for 75 studies. Upon full-text review, 21 studies were excluded, and 61 were retrieved and examined for eligibility. Thirty-one studies were included in the final evaluation. The reasons for exclusion at this point were an irrelevant study population *(n* = 6), inappropriate study design (*n* = 10), missing outcome data (*n* = 11), outcomes not specifically related to dementia or mild cognitive impairment (*n* = 6), or a lack of clearly stated conclusions (*n* = 11).

The PRISMA flow diagram illustrates this systematic process, ensuring that the final list of studies selected for review met predefined inclusion and exclusion criteria, thereby enhancing the validity and reliability of the findings, as shown in Figure 2.

### 2.5. Quality Assessment (Risk of Bias)

The assessment of risk of bias and methodological quality was conducted using the Joanna Briggs Institute (JBI) Critical Appraisal Checklists, which were selected based on the study design (e.g., randomized controlled trials, cross-sectional studies, qualitative research, mixed-methods studies, and systematic reviews) [36]. The JBI consists of several study questions, resulting in an overall appraisal decision for (1) inclusion, (2) exclusion, or (3) seeking further information (Appendix A). Only studies which met the minimum quality threshold for inclusion were retained.

The results of the assessment were used to inform data extraction and synthesis, as well as to guide the overall evaluation of each study’s contribution to the evidence base. Two reviewers (AA and MGR) independently conducted the quality assessments. Disagreements were then resolved through discussion, and when consensus could not be reached, a senior reviewer (VP) was consulted to resolve discrepancies. For each study, the JBI score was calculated by dividing the number of “Yes” responses by the total number of applicable checklist items (excluding “Not Applicable” responses) and then converting the result into a percentage. In the next stage, studies were categorized based on identified JBI scores, as follows: high quality (≥75%), moderate quality (50–74%), and low quality (<50%) [36,37].

Among the 31 included studies, 27 (87%) were rated as high-quality, while 4 (13%) were rated as moderate-quality. This distribution generally indicates a high level of methodological rigor, with some variability. High-quality studies frequently reported clearly defined objectives, appropriate and transparent study designs, and the use of valid and reliable outcome measures. In contrast, moderate-quality studies often lacked detail regarding sampling strategies, data analysis procedures, or reporting of study limitations.

## 3. Analytical Process

The systematic review and meta-analysis (using Stata 18) revealed a relationship between greater ownership and utilization of ICT devices and better cognitive reserve, accompanied by less subjective cognitive distress, among older adults. Most studies have shown that the repetitive use of electronic devices (such as tablets and mobile phones) helps improve memory and executive function and protects against cognitive decline [38].

The final study investigated the trends in the use of ICT devices among older people with dementia. There is a remarkable rise in the number of papers published in 2022 regarding ICT usages and interventions, and this peaked in 2024, indicating that there is increased research interest in using ICTs in cognitive health management. This shows an increasing awareness of technology in dementia care, accompanied by growing research evidence supporting the development of new assistive technologies and e-healthcare solutions.

### 3.1. Word Cloud Visualization of Key Terms

A word cloud (Figure 3) is a robust method for extracting knowledge from text corpora by representing term frequency in a weighted visual matrix. For this review, we examined the 38 abstracts of the systematic literature review (SLR) using atlas.ai, excluding general words such as “study,” “category,” and “topic,” as well as other stop words, to avoid missing significant content [39]. While some broad terms were repeated, the resulting map highlighted nuanced yet essential differences, emphasizing the field’s focus on cognitive, digital, and usability issues in interventions for older adults with mild cognitive impairment or dementia.

### 3.2. Key Findings

Figure 4 illustrates the publication trend from 2015 to 2025, which indicates a growing academic interest in the adoption and appropriation of ICT devices among older adults. The line chart represents the number of publications from 2017 to 2025, with a fluctuating trend over the years. It was two in 2017, which dropped slightly to one in 2018. It followed a steady increase to three in 2020, four in 2021, and five in 2022. It fell to 2 in 2023 before jumping to 12 publications in 2024, which was the highest number of papers published during the period focused on in this study. In 2025, the number of publications decreased again to two. Overall, the data demonstrates an irregular growth pattern with two high peaks in 2022 and 2024. The rise in research in 2024 indicates a higher ICT adoption rate, increased digital literacy, and the potential to reach older people [40]. The pandemic accelerated the adoption of ICT among elderly people with dementia, enhancing their use of telemedicine, online banking, and social networking platforms. Studies on digital inclusion also explored this group, including the adoption barriers of this population. This smart home and health technology, which utilizes AI, has also triggered additional studies on the use of new technology among older people [41]. In 2025, two journal articles were recapitulated in a monthly timeline. The study reveals an increasing research interest in utilizing ICT in web activities, cognitive functions, and healthcare access.

Figure 5 illustrates the prevalence of study types in ICT adoption research among older adults. Cross-sectional studies (*n* = 10) predominate, focusing on the statistical examination of technology use, while cohort studies, quasi-experimental studies, and systematic review studies (*n* = 4) investigate users’ experiences and perceptions. Qualitative and randomized controlled studies each contributed (*n* = 3), demonstrating a balanced presence of evidence synthesis, longitudinal assessment, and intervention-based approaches. Additionally, narrative studies (*n* = 2) and mixed-method studies (*n* = 1) provide longitudinal insights, by synthesizing existing literature and providing critical perspectives on digital inclusion, usability challenges, and the role of assistive technologies in aging populations.

Figure 6 illustrates the countrywide distribution, where the largest group, “Other Countries,” comprises 14 studies, indicating several international contributions. The USA follows with 10 studies, while South Korea accounts for 5 articles, and Canada accounts for 2. The chart reflects a notable contribution from the USA, with significant contributions from other countries, reflecting higher but not equal global interest in ICT adoption research among older adults.

Figure 7 presents the distribution pattern of ICT devices used across studies targeting cognitive health interventions for older adults with mild cognitive impairment (MCI) and dementia. Among the various technologies, the highest percentage (32.3%) involved the use of ICT platforms, indicating a general adoption of digital tools. The second-largest category was tablet-based interventions, which were observed in 16.1% of studies, reflecting their popularity due to their ease of use and portability. Next, wearable devices accounted for 13.0%, and assistive technologies and smartphones each represented 9.7%, indicating balanced adoption across these user-friendly tools. Meanwhile, telephone-based interventions made up 6.4%. Less commonly employed tools included robotics/artificial intelligence (AI) assistive technologies, mHealth/eHealth tools, and computers, each comprising 3.2% of the total. Furthermore, other technologies were found to account for 3.2%, indicating moderate use of support equipment and miscellaneous, unclassified computerized solutions. Notably, very few respondents reported daily technology use, and those who did engaged with a diverse range of devices, suggesting an underutilization of familiar technologies within formal interventions. 

## 4. Results

Table 3 provides a summary of recent studies on the adoption and use of technology among older adults with dementia or cognitive impairment. It represents key study characteristics, including study design, study settings, device types, type of intervention, and findings on the usability, effectiveness, and accessibility of digital and cognitive effects tools for managing dementia.

### 4.1. ICT Intervention Types in Dementia Care

The following section outlines key categories of ICT interventions commonly employed in dementia care, highlighting the unique features, benefits, and limitations associated with each device type.

#### 4.1.1. ICT Platforms

ICT platforms and devices have also emerged as an essential assistant in the care of dementia patients due to their wide-ranging ability to deal with cognitive impairments and promote social interaction among older adults with dementia or MCI [42,45,46]. These technologies enhance daily life, mental stimulation, and social interaction, significantly improving the quality of life for individuals who age independently [42,46,53]. Studies have depicted significant cognitive improvement through interactive digital media that involves people in frequent cognitive stimulation [45]. Furthermore, ICT interventions virtually eliminate social isolation by providing frequent opportunities for social interaction, which benefits dementia patients living in isolated settings [46].

Empirical data consistently shows substantial improvements in cognitive and daily functional capacity among older adults who use these ICT platforms, which legitimize their inclusion within dementia care plans [46,49,50,62]. Likewise, social connectivity has emerged as a crucial protective factor for cognitive health; several studies have focused on using digital communication platforms to promote engagement [45,46]. Video conference applications, social media groups, community network sites, and online forums allow older adults to maintain interpersonal relationships, participate in group activities, and access support services despite their cognitive or physical limitations. The use of these platforms in daily life was associated with improved mood, reduced loneliness, and increased mental stimulation [45,46]. However, problems such as insufficient trust in online platforms, a lack of digital literacy, difficulty in managing passwords, and fears of online fraud were prevalent among older users [69]. Therefore, customized, tailored digital training programs and user-friendly platform designs were identified as acute facilitators for effective social ICT engagement among this population. Social interaction websites, such as video conferencing platforms and web chats, have helped alleviate loneliness and stimulate cognitive activity [49,50]. However, they have also presented issues of digital literacy and cybersecurity threats that require regular attention [45,46]. Innovative care services based on ICT have been explicitly highlighted as significantly improving physical and cognitive functionality, as well as addressing the loneliness and inactivity problems commonly experienced by older populations [55]. After experiencing initial use issues, the technologies provide significant long-term benefits, suggesting flexibility and sustained use [66,69].

#### 4.1.2. Tablet-Based Devices

Tablet-based interventions have become highly effective among older adults due to their simplicity, portability, interactive properties, and user-friendly interfaces. The general acceptability and user satisfaction of a tablet-based multitasking platform in Singapore are notable, as digital therapy is not only acceptable but also benefits from customization and training support to enhance user engagement [43,59]. Likewise, a tablet-based screening tool for cognitive impairment emphasizes reliability and practical use in primary care settings [47]. Additionally, tablet-based “Smart Brain” application significantly improved mental function and reduced depression symptoms among older adults with dementia [51]. Further supporting these outcomes, the study demonstrated that an ICT-based multicomponent program delivered via tablets effectively enhanced both body composition and cognitive performance [65,68]

#### 4.1.3. Assistive Technologies

Dementia-specific assistive technologies have also been crucial in enhancing the daily functioning, independence, and quality of life of individuals with dementia [59,60]. Numerous studies have reported that these interventions enhance users’ self-management, autonomy, and functional performance capacity, accompanied by high levels of user satisfaction and significant practical benefits [59,60]. Personalization and ethics are emphasized as crucial for the optimal effectiveness and usability of assistive technologies in providing personalized care for individuals with dementia [67]. This approach maximally enhances user experience and effectiveness in real-world dementia management.

#### 4.1.4. Wearable Devices

Wearable devices continuously track physical and mental health, providing timely and proactive interventions tailored to individuals’ needs. Wearables’ capability to perform continuous tracking enormously enhances their potential in preventive and personalized healthcare intervention among dementia patients [52,61]. Integrating wearable technology with exergaming further enhances these benefits, providing both cognitive and physical health benefits simultaneously in a fun and interactive manner, thereby supplementing the overall effectiveness of wearables in comprehensive dementia management interventions. Additionally, random Forest models achieved 92.8% accuracy in classifying people with Alzheimer’s disease separately from healthy controls using behavioral and physiological data. In comparison, Support Vector Machines (SVMs) achieved up to 88.5% accuracy in distinguishing mild cognitive impairment (MCI) cases from normal aging populations [54]. These algorithms are commonly embedded in AI-driven cognitive assessment platforms, enabling real-time monitoring, risk stratification, and adaptive feedback for older adults with MCI or dementia [54].

This Figure presents an AI-enabled cognitive care framework. It integrates three core technologies: AI-based cognitive assessment systems for structured testing, wearable devices for continuous physiological and behavioral monitoring, and tablet-based apps delivering interactive cognitive training (Figure 8). A centralized AI engine processes multimodal inputs to generate real-time, personalized interventions. The framework supports early detection, user engagement, and scalable delivery of tailored cognitive health solutions. It reflects key principles of digital precision health and is well-suited to support aging populations and dementia care in remote and community-based settings.

#### 4.1.5. Smartphone Devices

Smartphones offer enhanced cognitive and social interaction capabilities through various apps, which help reduce loneliness considerably and enable regular, high-quality social interactions [48]. Earlier studies also highlight the promising potential of telephones for substantially enhancing social activity, building formative evidence for long-term use and adoption in dementia care [71]. In addition, frequent use of smartphones among older adults was associated with fewer subjective cognitive concerns, particularly in executive functioning, reinforcing the role of everyday digital engagement in supporting cognitive well-being [57].

#### 4.1.6. Telephone

Telephone technology is the prime tool for facilitating social connectedness, psychological well-being, and mental stimulation among people with dementia [44,56]. Telephonic interventions are crucial for addressing the critical need for frequent social interaction and emotional support, particularly among older populations who are geographically remote or socially isolated [44].

#### 4.1.7. Robotics and AI Assistive Devices

Advanced robotic and AI-based technologies provide excellent cognitive stimulation and emotional support, offering highly interactive therapeutic benefits for individuals with dementia [63]. Robot-based interventions, particularly those involving humanoid robots, significantly enhance cognitive engagement, emotional interaction, and overall cognitive functioning, highlighting new prospects for active and engaging interventions for individuals with dementia [63].

#### 4.1.8. mHealth and eHealth Interventions

Both eHealth and mHealth interventions significantly contribute to improving the functional status, autonomy, and quality of life of patients with dementia [66]. Personalized visual mapping technologies and other digital solutions allow control over tasks of daily living, significantly enhancing independence and daily competence in older patients with dementia [66].

#### 4.1.9. Computer

A computer-based tele-exergame system that combines sensor-driven balance training with cognitive tasks for older adults with MCI or dementia [58]. Delivered via a telemedicine interface, the intervention allowed users to self-administer interactive exercises at home. The study demonstrated strong feasibility and acceptability, with participants reporting positive attitudes and improved cognitive function [58].

#### 4.1.10. Other Devices

Other notable technological interventions include computer-based applications, broad-spectrum, internet-enabled devices, and a variety of general, non-device-specific technological solutions [70]. Innovative technology solutions continue to expand in scope, offering innovative pathways for detecting, predicting, monitoring, and managing dementia. Several studies have provided a comprehensive overview of innovative systems ranging from sensor-based networks to AI-enabled monitoring platforms that contribute to real-time, data-driven dementia care strategies. These integrated approaches support early intervention and continuous health monitoring, emphasizing the growing relevance of ubiquitous computing in personalized dementia management.

Similarly, research has investigated the usability and acceptability of various technological solutions among community-dwelling older adults with mild cognitive impairment or dementia. Their findings underscore the value of involving older adults and caregivers in the design and implementation process to ensure that systems are effective and user-friendly. This review categorizes technologies by purpose, such as safe walking, independent living, and social engagement, and reinforces the need for ongoing participatory development to increase adoption and positive outcomes in dementia care [70].

### 4.2. ICT Adoption Enablers and Barriers

Table 4 presents a structured synthesis of adoption enablers and barriers across ten categories of ICT used in dementia care. The matrix reflects insights from 31 peer-reviewed studies, highlighting both transformative enablers and ongoing obstacles of ICT solutions for older adults with MCI or dementia.

#### 4.2.1. ICT Platforms

ICT platforms (such as web-based portals, videoconferencing systems, and virtual social spaces) excel in their cognitive stimulation capabilities [42,45,46,53] and ability to facilitate social interaction [47,50,70], promote their independence [50,62], and support remote healthcare delivery and services [55,69]. However, there are several significant barriers hindering the adoption of ICTs, including limited digital literacy [45,69], security concerns [46], and a lack of trust in platforms among older adults [53,69]. Additional barriers include poor password management, difficulty navigating portals with small fonts or confusing layouts, and inconsistent access to high-speed internet, which is vital for video functionality. Many users also express anxiety over teleconsultations due to a lack of privacy in shared living environments.

#### 4.2.2. Tablet

Tablets are widely accepted due to their interactive interfaces [43,47], intuitive usability [47,51], and portability [65,68], mainly when they are used for cognitive training or mental health applications. Despite these enabling factors, barriers persist, including training requirements [51], user fatigue [65], and limited customization [68], which continue to hinder sustained use. Older adults also cite small on-screen keyboards, difficulty adjusting brightness and contrast due to vision impairments, and a lack of physical dexterity support as additional challenges.

#### 4.2.3. Assistive Technologies

Assistive technologies (such as memory aids, medication reminders, and smart home tools) are recognized for promoting autonomy [59,60], supporting daily functioning [60], and enabling personalized care [67]. However, ethical concerns [68], limited personalization [59], and a lack of flexible design [60] demonstrate obstacles to broader integration. Privacy issues arise when sensors are placed in bedrooms or bathrooms. Furthermore, some systems generate excessive alerts or false alarms, which can be frustrating for both users and caregivers.

#### 4.2.4. Wearables

Wearables provide real-time, continuous monitoring of health and cognitive status [52,61], enabling early detection of decline [52] and supporting preventive strategies [72]. These benefits are often offset by discomfort [61], sensor inaccuracy [52], and concerns about privacy [72]. Additional challenges include frequent charging requirements, overly complex interfaces on companion apps, and concerns about GPS tracking being conducted by third parties without users’ consent.

#### 4.2.5. Smartphones

Smartphones are praised for enhancing social connectivity [48,71], facilitating communication [72], and supporting routine cognitive tasks [57]. Yet, barriers remain, including interface complexity [58], cybersecurity concerns [48], and low technology confidence among older adults [71]. Specific issues include app overload, difficulty managing updates, stress caused by spam or scam calls, accidental app deletions, vision and fine motor limitations, which further compound navigation difficulties.

#### 4.2.6. Telephone

Telephone-based interventions, though low-tech, are consistently cited for their accessibility [45] and emotional support value for isolated older adults [56]. Still, their limited interactivity restricts deeper cognitive or therapeutic engagement [44,56]. Additional barriers include hearing impairments, the absence of visual cues, and difficulty using automated phone trees.

#### 4.2.7. Robotics and AI

Robotics and AI-assisted systems provide engaging and emotionally intelligent therapeutic interventions [63]. Despite their potential, steep learning curves and user skepticism about AI remain serious deterrents to implementation [63]. Older adults report that robots can be perceived as impersonal, confusing, and intimidating at times. Some fear dependency on machines and express frustration when voice recognition fails due to dialects or speech impairments.

#### 4.2.8. mHealth and eHealth

mHealth and eHealth tools, such as mobile health apps and web-based health platforms, are shown to improve personalized care [66] and overall health outcomes [66]. Nonetheless, these devices often face issues related to a lack of awareness and usability issues that are primarily prevalent among older individuals [66]. Barriers include overly medical language in content, lack of bilingual or audio support, and complex login authentication processes. Many apps also lack offline functionality, which limits access for users in rural areas.

#### 4.2.9. Computer

Computer-based interventions, such as those used for home-based cognitive training and exercise (e.g., tele-exergames), have been shown to have positive effects on digital engagement and cognition [58]. However, they may present usability challenges due to maintenance requirements and complexity for older adults [58]. Specific issues include long loading times, accidental file deletion, and difficulty managing device settings or updates. Caregivers often need to provide recurring technical support.

#### 4.2.10. Other ICT

Other ICT applications, such as general-purpose sensor networks and integration tools, offer a broad spectrum of utility for detection, monitoring, and care coordination [70]. The primary challenges of this group stem from poor personalization and a lack of user involvement in design processes [70]. Additionally, many of these tools are integrated into commercial packages without transparency in pricing, raising concerns about their affordability. A lack of multilingual interface options and a lack of cultural relevance also hinder their uptake among diverse aging populations.

### 4.3. Effectiveness of ICT-Based Interventions Regarding Cognitive Outcomes

The forest plot displayed (Figure 9) presents the standardized mean differences (Cohen’s *d*) from 11 studies that estimate the effect of ICT-based interventions on cognitive outcomes among older adults. Each study’s effect size is denoted by a square, with horizontal lines indicating the 95% confidence interval (CI). The pooled effect size, represented by a red diamond at the top, is 0.59 [0.14, 1.03], indicating a moderate and statistically significant overall effect.

Several individual studies, including those by Kim et al. [55] (*d* = 0.43, CI: [0.13, 0.73]), Coley et al. [62] (*d* = 0.49, CI: [0.08, 0.90]), and Choi et al. [50] (*d* = 0.37, CI: [0.02, 0.72]), reported significant small-to-moderate effects. Other studies, such as those by Malinowsky et al. [72] and Chae & Lee [51], demonstrated larger effect sizes (*d* = 1.07 and 1.18, respectively), although their wider confidence intervals (CIs) indicate variability and possible heterogeneity. Some studies have reported large effect sizes, such as those by Malinowsky et al. [72] (*d* = 1.07) and Chae & Lee [51] (*d* = 1.18), with wider confidence intervals, indicating greater uncertainty and potential heterogeneity.

In contrast, Kim et al. [55] and Park et al. [58] reported non-significant effects with confidence intervals crossing zero, suggesting greater uncertainty in the intervention effect. Finally, the pooled effect size of 0.59 indicates that ICT-based interventions exert a moderate positive influence on cognitive function in older adults. While there is some variation across these studies, the direction of effect is consistently positive, and the confidence interval does not include zero, indicating statistical significance. These findings collectively support the efficacy of ICT tools in enhancing cognitive health and mitigating age-related cognitive decline.

For further analysis of the sources of heterogeneity observed in the meta-analysis, the researchers also conducted a structured subgroup analysis, categorizing studies by the mode of intervention (such as tablets, ICT platforms, wearable devices, assistive technology, computers, and smartphones), as shown in Figure 10. This subgroup analysis uncovered variability in effect sizes across different intervention types, with tablet-based interventions generally yielding larger effect sizes (e.g., Chae & Lee [51], *d* = 1.18 and Cay et al. [52], *d* = 0.87) compared to ICT platforms and other modalities, which demonstrated more minor, but still positive effects.

Heterogeneity, as measured by I^2^, increased from 46% to 55% after excluding high-risk studies, indicating that the removal of lower-quality studies may have overstated the variability among the remaining studies. This suggests that study quality and intervention type are essential determinants of heterogeneity. Meta-regression for the severity of dementia was not possible due to the paucity of reporting dementia severity in studies, but this is a critical area for future research.

This means that study quality and intervention type are essential predictors of heterogeneity. Unfortunately, meta-regression by severity of dementia could not be undertaken, since reporting across studies of severity of dementia was poor, but this is something that could be addressed in future studies.

### 4.4. Heterogeneity and Publication Bias Assessment

The funnel plot illustrates the reliability and robustness of the meta-analysis findings, where the studies (Figure 11) provide a visual analysis of publication bias by plotting effect sizes (Cohen’s *d*) against their standard errors. Although the overall distribution appears relatively symmetrical, minor asymmetries are present, particularly at the lower-precision end. This asymmetry suggests publication bias because very few studies with non-significant effects appear to be underrepresented. Additionally, the lack of studies in the lower-left quadrant of the plot, along with the concentration of small studies that report moderate-to-significant positive effects, has likely inflated the overall pooled estimate. The trend is suspicious, suggesting that more positive results were published and considered, while null results may have been excluded. As a result, the meta-analysis slightly overestimates the efficacy of ICT-based interventions regarding cognitive and behavioral outcomes.

Table 5 shows that Egger’s regression test indicated significant funnel plot asymmetry, with an intercept of 1.83 (SE = 0.30; 95% CI: 1.16 to 2.51; Z = 6.14; *p* < 0.001) and a negative slope of −0.25 (SE = 0.12; 95% CI: −0.49 to −0.01; *p* = 0.04), suggesting the presence of small-study effects. To further assess potential bias, we performed a Duval and Tweedie trim-and-fill analysis. The unadjusted pooled effect size was Cohen’s *d* = 0.46 (95% CI: 0.35 to 0.57), which decreased slightly to *d* = 0.39 (95% CI: 0.28 to 0.49) after adjusting for missing studies.

### 4.5. Interpretation of Heterogeneity (I^2^)

The overall heterogeneity among the 11 studies included in the meta-analysis was found to be moderate (I^2^ = 46.0%), indicating some variability in the study outcomes. After excluding two high-risk studies (Kim et al. [55] and Choi et al. [50]), the heterogeneity increased slightly (I^2^ = 55.4%), indicating substantial heterogeneity. This level of heterogeneity reflects differences in intervention modalities, population characteristics, and study design quality across the included studies.

### 4.6. Sensitivity Analyses

To examine the robustness of the meta-analytic findings, we employed a leave-one-out approach, and sensitivity analyses were conducted by excluding studies considered to have a high risk of bias. The overall meta-analysis, which included all studies, yielded a pooled effect size of Cohen’s *d* = 0.49 with moderate heterogeneity (I^2^ = 46.0%). After eliminating two high-risk studies (Kim et al. [55]; Choi et al. [50]), the effect size increased only slightly to *d* = 0.50, while heterogeneity rose slightly to I^2^ = 55.4%. This minor variation in the pooled effect size suggests that the results are stable and not unduly influenced by the presence of high-risk studies, thereby reinforcing the reliability and robustness of the overall findings (Table 6).

## 5. Discussion

This systematic review revealed the increasing importance of ICT-based interventions in facilitating cognitive ability and mental well-being of older adults with MCI and dementia. A wide range of devices, including wearable sensors, AI platforms, smartphones, and tablets, promise cognitive stimulation, early detection of deteriorating cognitive function, and facilitation of social activity. This systematic review summarizes the key findings regarding the contribution of ICT devices to dementia care and cognitive functioning in older people (Figure 12).

The COVID-19 pandemic accelerated the quick normalization and adoption of ICT-based interventions among older adults, especially in the areas of telehealth, mHealth/eHealth, and social technologies. Virtual check-ins, remote care models, and digital social platforms became crucial for maintaining cognitive engagement and access to services [42,44]. This shift increased digital inclusion among populations that were previously underserved by technology, although disparities in access and literacy persisted [45,46]. Studies from 2022 to 2023 demonstrate the increasing prevalence and acceptance of caregiver-mediated digital tools [45,46], remote cognitive screening [56], and mobile health monitoring [56] following the pandemic. Moreover, pandemic-induced restrictions highlighted the importance of flexible, remote health support systems, prompting innovations in usability, personalization, and caregiver engagement [46]. These changes have prompted a reevaluation of the role of ICT in resilient aging care infrastructure.

Notably, smartphones and tablets were ubiquitous and adept venues for cognitive training and communication; however, a chief hindrance to equal access was the digital literacy gap. Wearable technology was the technological leader in facilitating passive monitoring of cognition and health status in everyday settings, but maintenance requirements and privacy threats deterred its traction. The intersection of AI technologies has brought together personalization and prediction capabilities, but it has also raised new ethical concerns, including concerns about data transparency and algorithmic fairness. By contrast, tele-exergames and robot systems can effectively combine physical and mental exercise, but they face hurdles in terms of cost and usability complexity.

Despite the promise of these technologies, digital exclusion remains a reality. Financial, infrastructural, and educational obstacles disproportionately affect minority and low-income older adult populations. Consistent with previous research [73,74], interventions must focus on user-centered design, digital literacy assistance, and culturally tailored adjustments to make technological advancements accessible to all individuals at risk of cognitive decline. Furthermore, the temporal development of ICT interventions from initial positions as plain communication centers into sophisticated prediction and monitoring systems reflect enormous technological changes. Still, it must also evolve in response to similar interests in ethical standards, policy frameworks, and caregiver training. The long-term effectiveness of multimodal, integrative ICT solutions that combine cognitive support, health surveillance, social activity, and individualized risk avoidance in a unified, easy-to-navigate platform warrants further investigation.

While several ICT interventions report promising short-term outcomes, a significant gap remains in understanding the long-term sustainability of ICT usage. Few studies are constrained by focusing on follow-up durations (23% of the included studies provided follow-up data beyond six months, highlighting a significant gap in understanding long-term sustainability), pilot-scale implementations, or inadequate funding, which limits their ability to evaluate long-term impacts [51,52]. This lack of extended assessment hampers our understanding of how sustained ICT engagement affects cognitive, physical, and emotional health among older adults. Furthermore, the absence of structured integration into routine care workflows reduces the potential scalability of these technologies [59,61].

To bridge this gap, future research should implement long-duration, randomized controlled trials (RCTs) [64], conduct post-implementation follow-ups, and conduct real-world evidence studies [60]. These strategies would enable a greater understanding of cost-effectiveness, user engagement, and the evolution of interventions over time [64]. Continual monitoring will also provide valuable insights into the extent to which technology fatigue, changing user requirements, or caregiver reliance affect the long-term uptake of ICT among aging populations. Although numerous interventions were possible and initially practical, long-term compliance, impacts on various cultural and socioeconomic populations, and scaling ICT innovations to real-world community settings are gaps that need to be addressed. Addressing these gaps will be crucial to realizing ICT’s full potential for enhancing the quality of life and cognitive capacity of older adults globally. Therefore, Table 7 summarizes the key distinctions between this research and the previous research.

### 5.1. Ethical Considerations and Implications

With the increasing adoption of ICT-based interventions in dementia care and ethical concerns regarding data privacy, informed consent, autonomy, and equity of access, it is essential to establish digital interventions in an ethically configured manner. To achieve this, effective policy mechanisms are needed to advance cybersecurity, data protection, and accountability. Policymakers must enact congruent legislation to govern the ethical application of AI, telehealth, and remote monitoring technology in dementia care, a field that is becoming increasingly relevant. AI systems and remote monitoring technologies raise advanced questions regarding transparency, algorithmic justice, and user control over personal health information [59,67]. In addition, developing digital interventions ethically requires robust policy frameworks that prioritize data protection, cybersecurity, and accountability. Policymakers must establish harmonized legislation to govern the ethical use of AI, telehealth, and remote monitoring technologies in dementia care [67,69].

This review’s findings also include several actionable implications for care settings and policy-level adoption. First, ICT skills training should be integrated into caregiver and geriatric care education to enhance frontline workers’ digital competency and confidence [62]. Second, subsidies for devices and the internet should be provided under community health programs to address issues related to affordability, particularly for vulnerable populations [50,54]. Third, the design of intervention needs to prioritize participatory, user-centered methods that are culturally reflective and address the personalization requirements of diverse aging populations [67]. At the policy level, the more widespread implementation can be enabled by incentives for age-friendly ICT design innovation [70], public–private partnerships for investment in infrastructure [71], and policies that enforce accessibility and fairness in digital health access [60]. Robust guidelines must be formulated to achieve user control, safeguard vulnerable groups, and promote ethical innovation. Policies must also ensure parity of reimbursement and access to digital therapeutics so that ICT-based cognitive interventions become mainstream in formal health systems and are accessible to different patient populations [71]. Otherwise, the technologies will reinforce current health inequities rather than mitigate them.

### 5.2. Policy Direction

Policy interventions are necessary to overcome structural barriers to the adoption of ICT in dementia care. Actions must target growing digital literacy training programs for older individuals and carers, subsidizing the acquisition of devices and broadband services, and technologically appropriate technical support [42,62]. Overcoming cost barriers and rural access inequalities is critical to achieving equitable uptake of these technologies [46]. Furthermore, the government must implement rigorous regulatory measures to protect data, ensure privacy, and promote the ethical use of AI-based platforms [67,69]. Digital health tools must be integrated into national health policies, including sufficient reimbursement mechanisms and access guarantees, to facilitate the large-scale implementation and sustainability of ICT-based cognitive interventions.

Recent economic evaluations indicate that investing in cognitive-supportive ICT is not only socially beneficial but also financially prudent. A 2024 review found that most technology-based interventions, such as telecare, wearables, and apps, are cost-effective when used as a form of dementia care [75]. For instance, Maintenance Cognitive Stimulation Therapy (MCST), a digital training tool designed to enhance daily cognitive engagement, was identified as particularly cost-effective, yielding meaningful improvements in cognitive function at a relatively low cost [76]. Likewise, supportive care models (cognitive therapy, caregiver training, and multicomponent interventions) resulted in significant health system savings [77]. Notably, a rural digital literacy program in South Korea reported measurable social returns, including improved well-being (+3.7 points) and better cognition (+1.1 points), which suggests that subsidy-funded training can deliver psychosocial dividends [78]. Supporting these initiatives with broadband investments, subsidized devices, and training may thus yield proportional reductions in health and social care expenditures, while also reducing equity gaps.

## 6. Future Research

Although encouraging initial results exist, significant gaps remain that future research must address.

Large-scale, long-term follow-up studies are necessary to compare the long-term cognitive, affective, and functional impact of digital interventions [66]. Multi-stakeholder funding models integrate research grants with investments in the healthcare system.Future trials must examine the impact of early prolonged digital activity on the postponement of cognitive decline onset and progression of dementia [57].Evaluations of the cost-effectiveness, feasibility, and real-world utility of customized digital interventions, remote cognitive testing, and serious games are necessary to optimize intervention designs [43,60]. Therefore, it is necessary to integrate the collection of real-world evidence through routine healthcare data.Moreover, it is essential to extend research into heterogeneous cultural and socioeconomic populations to strengthen global applicability and inclusiveness [42,69].Emerging technologies, such as voice-controlled assistants, socially assistive robots, and cognitive prediction tools based on AI, must be extensively validated for usability, ethical integration, and clinical utility [67].Future research should prioritize pragmatic trial designs with embedded sustainability assessments and stakeholder engagement throughout the implementation process.

## 7. Strengths and Limitations

To provide a more straightforward overview of our review’s methodological context, Table 8 presents the key limitations and strengths of this study in a simplified and easy-to-understand format.

## 8. Recommendation

Based on the systematic review and the framework shown in Figure 13, the following recommendations are provided to guide the integration of ICT interventions into dementia care.

Step 1. Assess Needs and Involve Caregivers

Educational strategies require systematic and continuous development to engage caregivers and older adults, promoting the acceptability of technology and enabling the integration of technology-enabled activities into daily life [18]. Caregiver involvement is necessary for both the initial adoption and long-term maintenance of ICT-based interventions.

Step 2. Tailored Interventions

ICT interventions should be advanced to accommodate people with dementia’s cognitive, physical, and emotional needs. Adapted AI-facilitated cognitive training is a promising approach to enhancing intervention quality and warrants further examination in practice [79].

Step 3. Educating Older Adults and Training Caregivers

Formal training should include the aging population and their carers to ensure maximum digital literacy for enhanced exploitation and use of ICT equipment [80]. Training would be necessary to focus not only on working with machinery but also on self-confidence and problem-solving abilities.

Step 4. Incorporate ICT into Daily Practice

Incorporation of ICT into daily life activities can also enhance independence and quality of life in people with dementia. Empirical evidence demonstrates that assistive technology, when incorporated into day-to-day practices without challenge, greatly improves functional outcomes and accelerates the development of person-centered care environments [59].

Step 5. Personalize ICT Applications

Technology design must be sensitive to the cognitive functions of older adults. Interfaces must incorporate large print, voice control, minimal navigation, and multifunctionality to improve usability and lower cognitive burden [81]. Individualized adaptations enhance patient satisfaction and medication adherence.

Step 6. Enhance Policy and Broadband Access

Policy intervention is urgently needed to bridge the digital divide. The broadband infrastructures must be subsidized, and the internet must be made affordable, with subsidies for ICT devices, particularly for low-income people and rural areas [42]. Inclusive deployment strategies are necessary to guarantee equitable access to digital dementia care innovations.

To mitigate digital disparities, which necessitate policy intervention in the short term, government investment in expanding broadband infrastructure, with a focus on making internet access affordable and supporting ICT devices for low-income and underserved communities, is essential [42]. There is a need for inclusive deployment strategies to ensure that innovations in digital dementia care are accessible to all.

Step 7. Sustain Use and Support Adaptation

Sustainable ICT adoption of interventions relies on ongoing digital skill development, community reinforcement programs, and ethical governance principles. As AI-powered solutions become increasingly prominent in dementia care [67], concerns regarding data confidentiality, patient autonomy, and platform transparency must be addressed. Sustainability initiatives should involve systematic digital skills refreshments and technology sustenance for older adults.

## 9. Conclusions

This systematic review and meta-analysis, conducted in accordance with the PRISMA guidelines, further strengthens the emerging role of ICT-supported interventions in enhancing cognitive function, independence, and social engagement in older adults with mild cognitive impairment (MCI) and dementia. All digital interventions have been proven effective in improving memory, executive function, well-being, and reducing symptoms of depression, regardless of the device used, including tablets, smartphones, wearables, AI applications, tele-exergame technology, or assistive robots [43,59,67].

Despite these advancements, barriers such as digital learning, usability, socioeconomic disparities, infrastructure gaps, and ethical concerns regarding data privacy and transparency continue to pose hurdles, which contribute to the slow adoption [42,46,49,69]. Furthermore, the current evidence base is limited by heterogeneity in study designs, a lack of long-term outcome data, and inadequate representation of diverse communities.

Interventions would need to adopt user-centered design approaches, address digital literacy gaps through training for caregivers and patients, and implement an equitable approach through policy improvements to achieve the maximum penetration of ICT in dementia care. Additionally, ethical standards must be integrated into the development and use of AI-driven technologies to ensure user independence and confidentiality [67,71].

Lastly, future research is required, particularly in the form of realistic and culturally compliant longitudinal studies, to prevent potential ICT applications for dementia from becoming mere technological interventions that are feasible, sustainable, accessible, and ethically responsible on a global scale. Such innovations have the potential to bridge gaps in ICT-first interventions, significantly promoting cognitive reserve, independence, and improved quality of life for millions of older adults worldwide.

## Figures and Tables

**Figure 1 healthcare-13-01421-f001:**
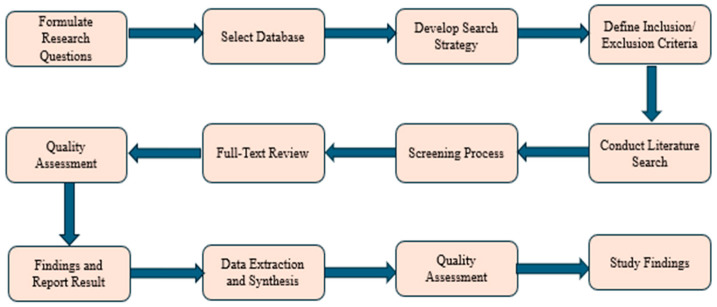
Systematic literature review (SLR) process for article selection.

**Figure 2 healthcare-13-01421-f002:**
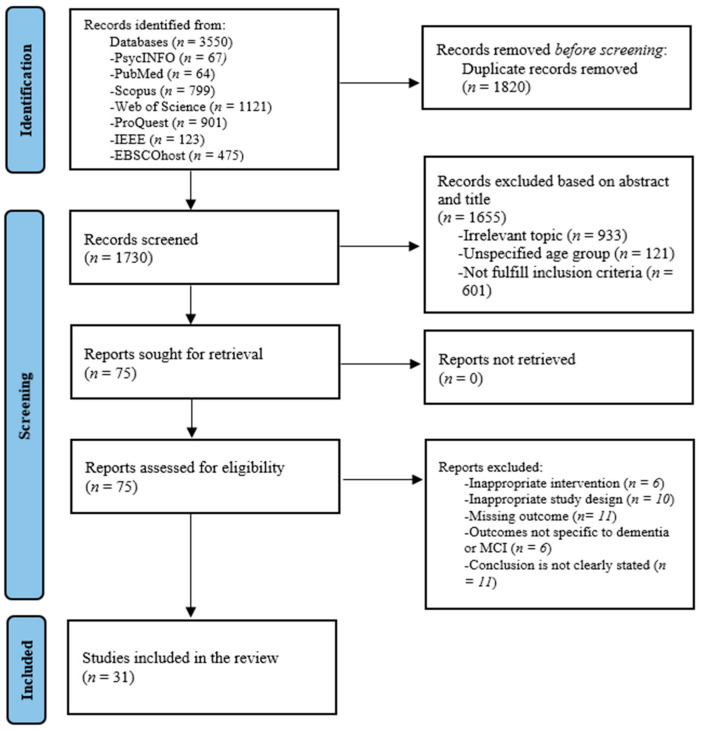
PRISMA flow diagram.

**Figure 3 healthcare-13-01421-f003:**
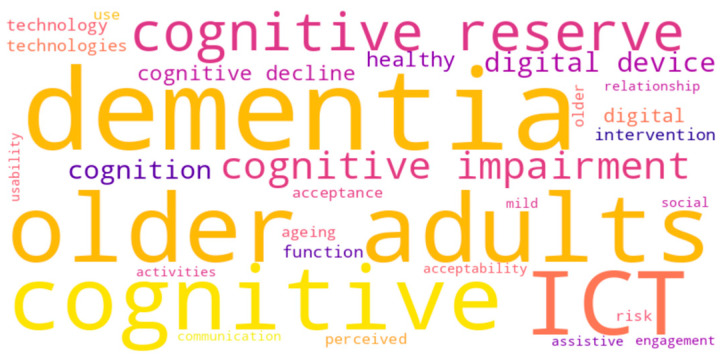
Word cloud visualization of key terms.

**Figure 4 healthcare-13-01421-f004:**
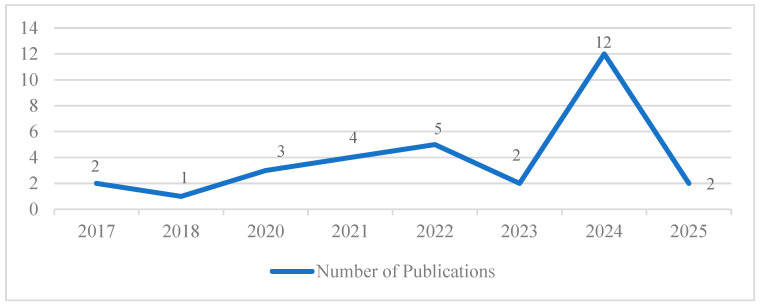
Yearly distribution of publications.

**Figure 5 healthcare-13-01421-f005:**
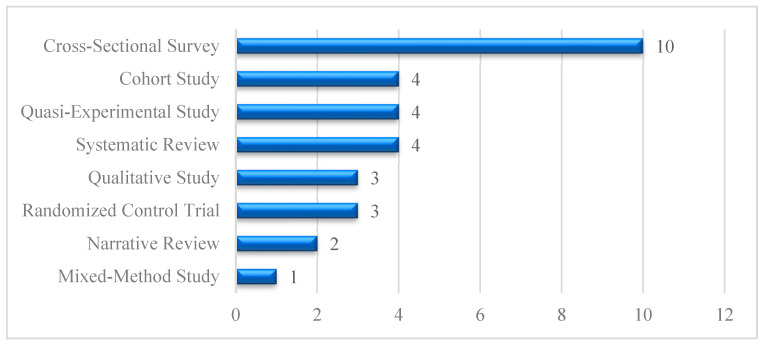
Distribution of study designs.

**Figure 6 healthcare-13-01421-f006:**
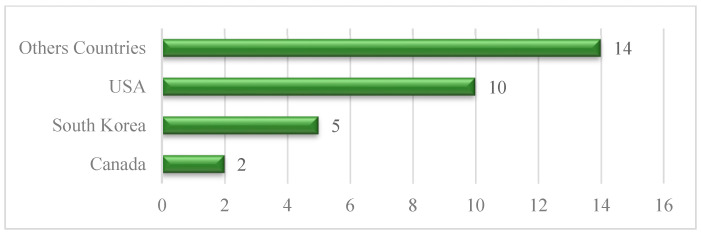
Geographical distribution.

**Figure 7 healthcare-13-01421-f007:**
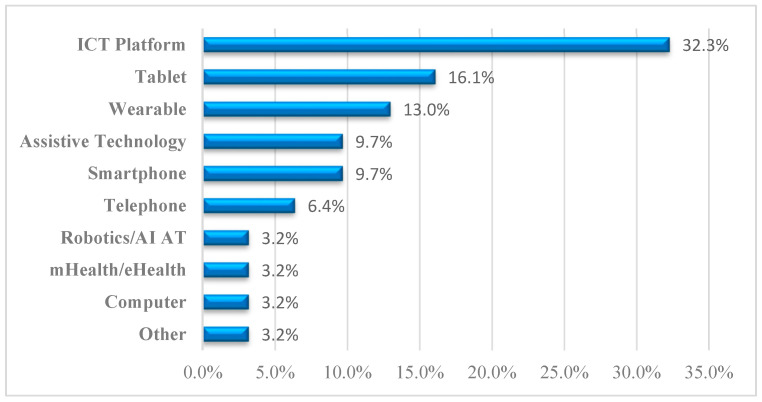
Distribution of device types used in ICT-based interventions for older adults with cognitive impairment.

**Figure 8 healthcare-13-01421-f008:**
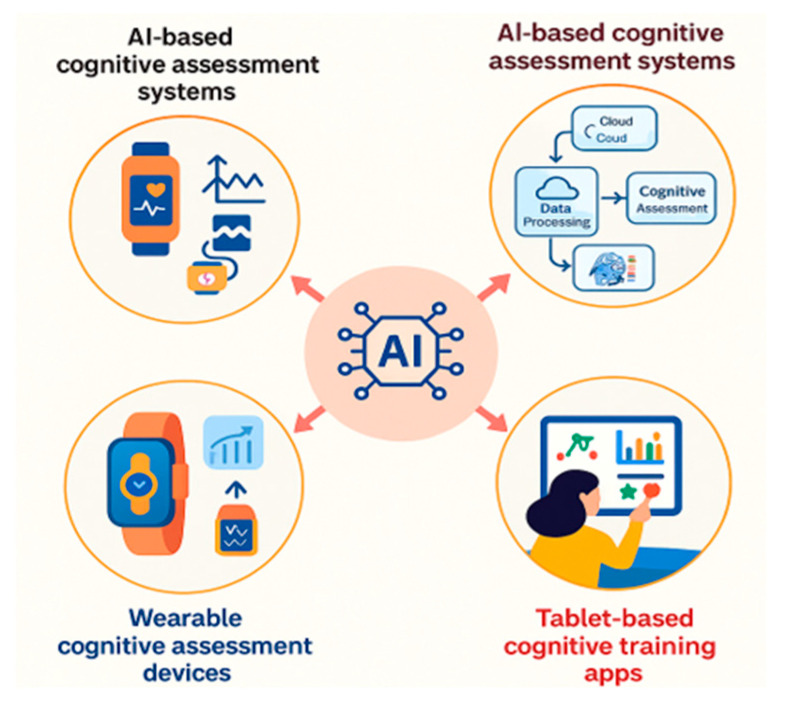
AI-integrated cognitive assessment and training systems.

**Figure 9 healthcare-13-01421-f009:**
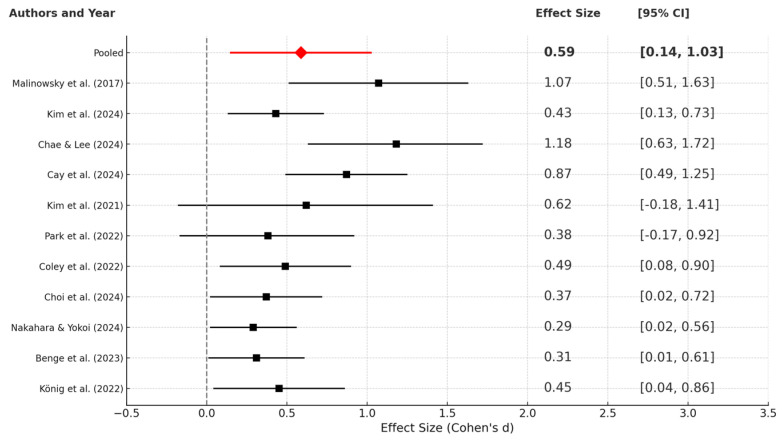
Forest plot showing random effects [46,50,51,52,55,57,58,59,62,64,72].

**Figure 10 healthcare-13-01421-f010:**
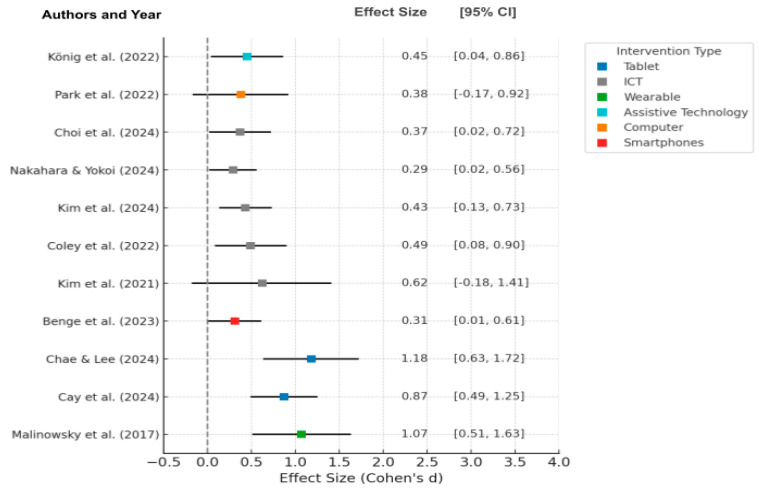
Forest plot of effect sizes (Cohen’s *d*) by intervention type [46,50,51,52,55,57,58,59,62,64,72].

**Figure 11 healthcare-13-01421-f011:**
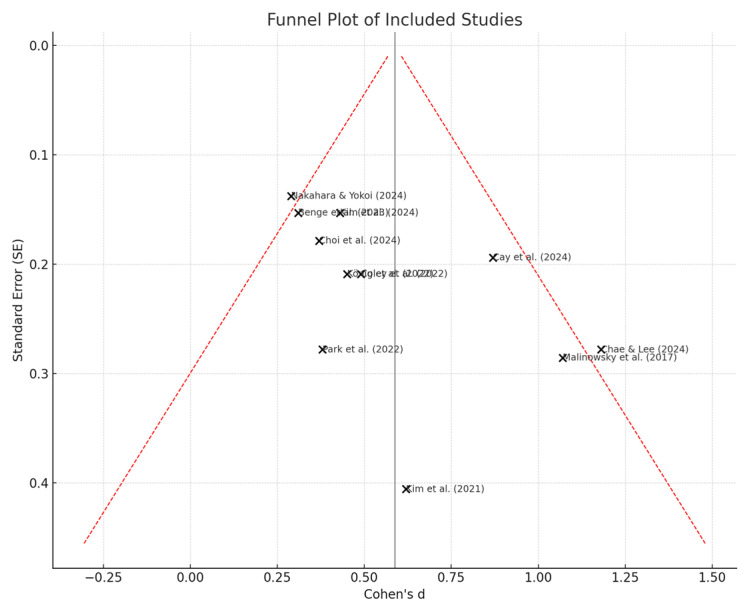
Funnel plot of meta-analysis [46,50,51,52,55,57,58,59,62,64,72].

**Figure 12 healthcare-13-01421-f012:**
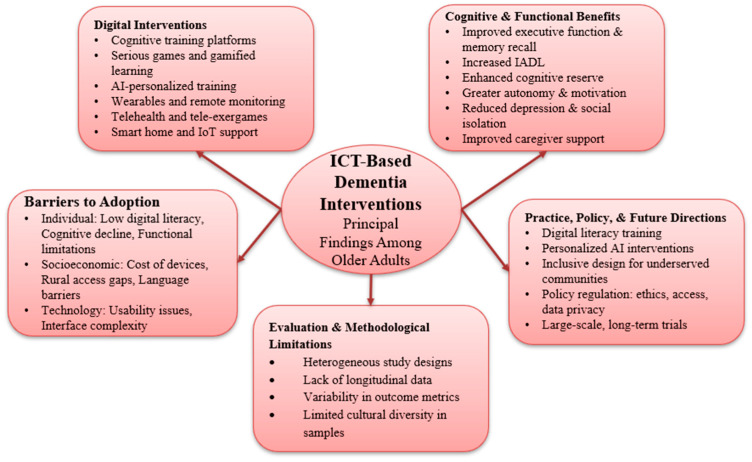
Overview of principal findings: ICT-based dementia interventions among older adults.

**Figure 13 healthcare-13-01421-f013:**
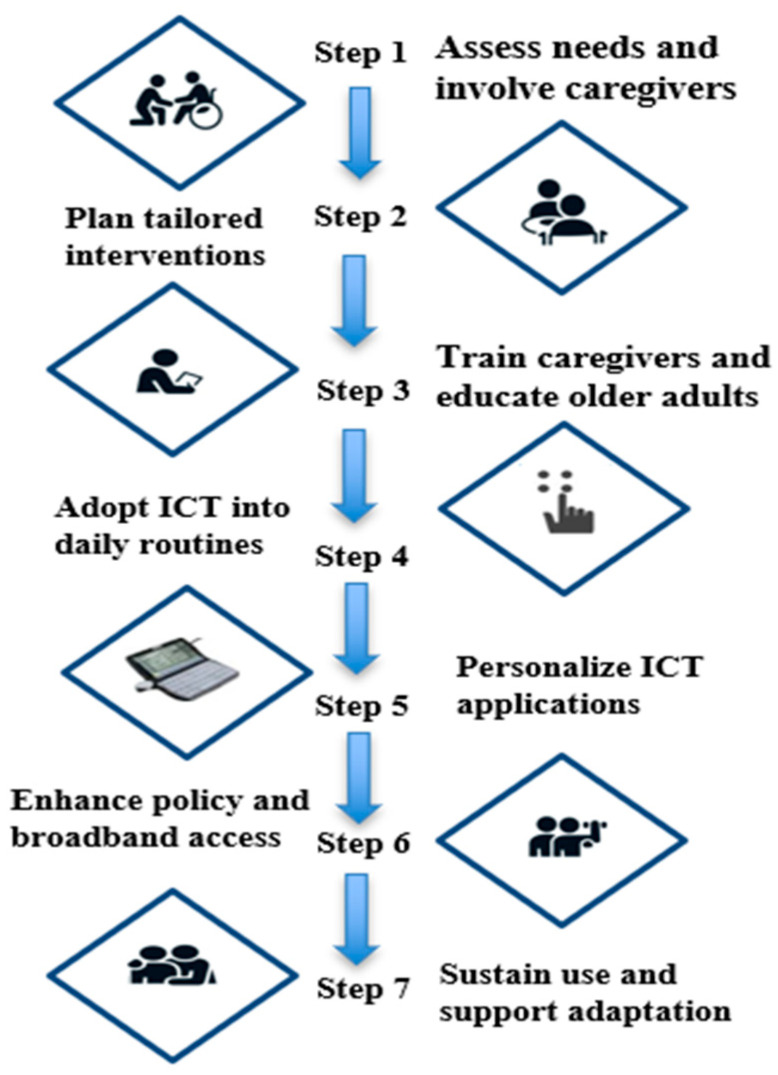
Framework for Integrating ICT into Dementia Care.

**Table 1 healthcare-13-01421-t001:** Inclusion and exclusion criteria (based on PECOS framework).

Criteria	Inclusion	Exclusion
**Population**	Older adults (≥55 years) are diagnosed with dementia, MCI, Alzheimer’s disease, or other neurocognitive disorders Caregivers facilitate digital intervention use	Adults < 55 years of age Studies examining ICT use in healthcare without a dementia/cognitive focus
**Exposure**	ICT-based interventions for dementia care, including - Digital cognitive training websites (e.g., *Smart Brain*, *CURATE.DTx*) - Telehealth and remote monitoring - AI-based diagnostic tools - Assistive technology (e.g., GPS tracking, wearables, smart homes) - mHealth apps and serious games - Efficacy studies focused on these technologies	Pharmacologic interventions Dietary modifications Non-digital assistive technologies Conventional caregiving methods Any other interventions not focusing on ICT-based cognitive support
**Comparator**	Usual care (non-digital dementia care) Non-digital cognitive training (e.g., paper-based memory tasks) No intervention (control groups) Other digital interventions	No relevant comparator (i.e., not assessing or contrasting ICT-based vs. non-digital approaches)
**Outcome**	**Primary Outcomes:** - Cognitive function (e.g., delayed recall, memory recall, executive function) - Behavioral and cognitive symptoms of dementia (BPSD) such as depression and anxiety - Activities of daily living (ADLs/iADLs) - Social engagement via technology-mediated interaction **Secondary Outcomes**: - Acceptability, usability, and adherence to ICT-based interventions	Not focusing on ICT’s effects on cognitive function, memory, quality of life (QoL), or ADLs/iADLs Irrelevant or unspecified outcomes
**Study Design**	Randomized controlled trials (RCTs), cross-sectional studies, cohort studies, mixed-methods studies, and systematic reviews Qualitative, quantitative, and literature review studies Peer-reviewed publications published between January 2015 and January 2025 Evaluations of ICT-based interventions for dementia care	Editorials, conference abstracts (insufficient data) Case studies (weak statistical power) Non-peer-reviewed preprints Publications published before 2015 (historical significance only)

**Note:** ICT = Information and Communication Technology; ADLs = activities of daily living; iADLs = instrumental activities of daily living.

**Table 2 healthcare-13-01421-t002:** Search terms applied in the SLR procedure.

Search Term	Databases	Results
**(“ICT” OR “Digital Health” OR “mHealth” OR “eHealth” OR “Telehealth” OR “Telemedicine” OR “Wearable Technology” OR “Mobile Health Applications” OR “Device Ownership” OR “Technology Adoption” OR “Digital Literacy”)** **AND** **(“Dementia” OR “Cognitive Impairment” OR “Cognitive Decline” OR “Alzheimer’s Disease” OR “Neurodegenerative Disorders” OR “Memory Loss”)** **AND** **(“Older Adults” OR “Aging Population” OR “Elderly” OR “Seniors” OR “Medicare Beneficiaries”))**	(a) PsycInfo (b) PubMed (c) Scopus (d) Web of Science (e) ProQuest (f) IEEE (g) EBSCOhost	67 64 799 1121 901 123 475

**Table 3 healthcare-13-01421-t003:** Study Characteristics of the selected articles on Technology Use for Dementia/Cognitive Impairment.

Study	Study Year	Study Design	Country	Study Setting	Technology Type	Groups in the Study	Device Used	Type of Intervention and Study Type	Objective	Main Findings	Cognitive Effects
**Martinez et al.** [42]	2025	Cross-sectional study	USA	Home-based	ICT devices (Smartphones, tablets, computers, the internet, social media, or other applications)	Older Asian Americans in affordable housing	ICT Platforms/Devices	Training cognitive function with a computer game	To investigate the relationships between perceived usefulness (PU), perceived ease of use (PEOU), ICT use, and loneliness among low-income, older Asian Americans.	1. ICT acceptance and use reduce loneliness 2. Ease of use influences ICT adoption 3. Digital literacy training is essential. 4. Policy support is needed for access to technology	Subjective cognitive decline
**Kwek et al.** [43]	2025	Qualitative study	Singapore	Lab-based playtesting and interview sessions	Tablet-based CURATE.DTx system	Community-dwelling older adults	Tablet-based	Digital therapeutic cognitive training	To evaluate the acceptability and user experience of CURATE.DTx, a multitasking-based DTx platform for cognitive training	1. Digital therapy is acceptable. 2. Users need more training 3. Customization improves engagement 4. Future improvements should target accessibility	Improved attention, engagement, and executive functioning
**Yan et al.** [44]	2024	Cohort study	Canada	Remote preoperative assessments	Telemedicine-based cognitive assessment tools (Telephone Montreal Cognitive Assessment)	Patients were assessed using four cognitive screening tools	Telephone	Remote cognitive screening tools are used through telemedicine	To determine the prevalence of suspected cognitive impairment using multiple screening tools	1. Screening tools detect cognitive impairment 2. Different tools yield varying prevalence rates 3. Remote assessments improve risk stratification. 4. Further validation needed	Detected impairment; varied by screening tool
**Peng et al.** [45]	2024	Cross-sectional study	USA	Data analysis from the National Health and Aging Trends Study (NHATS)	ICT devices, everyday technology, digital health technology	Homebound, semi-homebound, and non-homebound groups	ICT platforms/devices	Assessment of digital technology usage	To examine 1. The prevalence of digital technology use among community-dwelling older adults with or without homebound status 2. The association between digital technology use and homebound status.	1. Homebound seniors tend to have lower rates of technology adoption 2. Physical and cognitive limitations impact digital engagement 3. Accessibility remains a key barrier for homebound individuals.	Digital exclusion accelerates cognitive decline
**Nakahara and Yokoi** [46]	2024	Cross-sectional study	Japan	Community gathering places in Osaka Prefecture, Japan	ICT devices (including the internet, communication tools, etc.)	Older adults participating in social activities	ICT platforms/devices	ICT-facilitated social participation	To quantify how ICT use, participation frequency, and social networks influence cognitive function and loneliness among socially active people	1. ICT use strengthens social interaction, improving the quantity and quality of social participation among older adults 2. Frequent use of ICT improves cognitive function 3. ICT-driven engagement reduces feelings of loneliness	Improved cognitive function indirectly
**McMurray et al.** [47]	2024	Mixed-methods study	Canada	Primary care settings (FHTs)	Tablet-based digital screening tool (BrainFx SCREEN)	Older adults diagnosed with dementia	Tablet-based	Tablet-based cognitive impairment screening	To assess the validity, reliability, and applicability of the BrainFx SCREEN tool for MCI screening in a primary care context	1. Tablet-based screening shows promise 2. Accessibility improvements needed. 3. Large-scale testing is required 4. Integration with healthcare workflows is beneficial	Moderate sensitivity; limited reliability
**Hackett et al.** [48]	2024	Cross-sectional study	USA	Home-based and real-world mobility tracking	Smartphone with the mindLAMP app for passive GPS tracking	Community-dwelling older adults	Smartphones	Smartphone-based digital phenotyping	To assess the feasibility, acceptability, and validity of using smartphone-based GPS tracking to infer cognition, function, and mood	1. Mobility patterns link to cognition 2. GPS-based tracking shows cognitive decline patterns 3. Future studies should refine model accuracy	Greater mobility is linked to cognition
**Martinez et al.** [49]	2024	Cross-sectional study	USA	Survey-based data collection	ICTs (general technology use, internet, computers, mobile devices)	Low-income Asian American older adults	ICT platforms/devices	Technology acceptance modeling	To examine the role of self-rated health and subjective cognitive decline in ICT use	1. ICT use is linked to self-rated health perceptions 2. Digital skills training improves adoption 3. Socioeconomic factors influence engagement 4. Tailored policies can improve participation	Subjective decline moderated ICT use.
**Choi et al.** [50]	2024	Cross-sectional study	USA	Survey-based data collection	ICT and ICT-based communication and internet access (focused on cellphones, email, texting, and the internet)	Homebound and semi-homebound older adults	ICT platforms/devices	Assessment of the digital divide	To explore the digital divide between homebound and semi-homebound older adults using ICT device ownership and usage data	1. Homebound older adults face severe digital inequities. 2. Socioeconomic factors have a significant impact on access to ICT 3. Policy interventions can help bridge the gap 4. More targeted interventions are needed	Dementia is linked to reduced ICT use
**Chae and Lee** [51]	2024	Randomized controlled trial	South Korea	Participant’s home	Tablet-based training (Smart Brain program)	Community-dwelling older adults	Tablet-based	Digital Therapeutic Cognitive Training	To examine the effects of Smart Brain, an ICT-based cognitive training program, on multi-domain function in older adults with dementia	1. Cognitive function improved significantly in the intervention group. 2. Depression levels decreased among Smart Brain users. 3. Physical and nutritional health showed positive changes. 4. Participants reported high adherence and satisfaction.	Improved cognition, mood, physical, and nutritional status
**Cay et al.** [52]	2024	Cross-sectional study	USA	Lab-based reading task with speech recording	Wearable microphone to derive digital biomarkers (machine learning)	Cognitively impaired vs. cognitively intact groups	Wearable device	Speech-based digital biomarker analysis	To evaluate the effectiveness of speech-based digital biomarkers in detecting cognitive impairment severity	1. Speech biomarkers offer a viable screening method 2. Strong correlation with cognitive test results 3. May serve as an early diagnostic tool	Accurately detected and predicted cognitive impairment
**Anaraky et al.** [53]	2024	Cohort study	USA	Observational study using survey data	Internet, computers, tablets, texting, and emails	Community-dwelling older adults	ICT platforms/devices	Monitoring technology use for cognitive change	To determine whether technology use patterns could serve as an indicator of cognitive change in older adults	1. Tech usage changes can indicate early dementia 2. Monitoring online activity is useful 3. Declining engagement predicts cognitive issues	Technology discontinuation is linked to decline
**Addae et al.** [54]	2024	Narrative review	Various countries	Not specified (focuses on technological interventions)	IoT devices, wearable technologies, and machine learning algorithms	Older adults with dementia.	Wearable and IoT	Monitoring dementia	To explore smart and innovative solutions for early detection, prediction, monitoring, and management of dementia for the advancement of IOT	1. Wearable tech improves dementia management 2. AI-driven models help with early detection 3. Future research should integrate solutions	Supports early detection and monitoring
**Kim et al.** [55]	2024	Quasi-experimental study	South Korea	Older adults living alone	ICT-based smart care services for physical and cognitive functions	Older adults living alone in the community	ICT platforms/devices	ICT-based smart care services	To examine how ICT-based innovative care services affect physical and cognitive functions in older adults living alone	1. ICT-based innovative care services are effective at enhancing the physical function of the lower limb 2. Improvements observed in working memory and attention 3. Mixed results in global cognition (decreased K-MMSE score)	Improved working memory function
**Heponiemi et al.** [56]	2023	Cohort study	Finland	Community-dwelling adults	Internet use (not device-specific; focused on performance tests and self-reported digital use)	Community-dwelling adults	Telephone	Digital access competence prediction	To examine how impairments in visual, physical, and cognitive functioning predict internet use and digital competence	1. Older adults with physical and cognitive limitations are more likely to experience digital exclusion 2. Vision and physical functioning affect digital skill levels 3. Memory performance is a key indicator of digital competence	Smart technologies support mitigating cognitive decline.
**Benge et al.** [57]	2023	Cross-sectional study	USA	Observational research setting	Smartphones, social media, texting, and video calls	People having internet access	Smartphones and social media	Technology use and subjective cognition	To evaluate whether the frequency of digital device use is associated with greater or lesser subjective cognitive concerns (SCCs) in older adults. Cross-sectional study using hierarchical multiple	1. Increased device use is associated with fewer symptoms of cognitive control (SCC), especially in terms of executive function 2. General device usage matters more than use of social media or texting 3. Digital engagement may protect cognitive function with age	More use is linked to fewer concerns
**Park et al.** [58]	2022	Quasi-experimental study	USA	Sensor-based in-home interactive exercise system (tele-exergame)	Home-based remote intervention	Sensor-based tele-exergame system with a telemedicine interface	Computer	Telemedicine-based exergame training	To assess the feasibility, acceptability, and effectiveness of a sensor-based in-home tele-Exergame system for cognitive and motor function improvement in older adults with MCI/dementia	1. Tele-exergames are feasible. 2. Engagement was high 3. The long-term impact requires further study 4. Technology accessibility needs improvement	Cognition improved with the tele-exergame
**König et al.** [59]	2022	Qualitative study	Various countries	Home-based intervention	MEMENTO, a system of two e-ink tablets and a smartwatch	EG assistive intervention with MEMENTO	Assistive Technology	Assistive digital device system	To evaluate the usability, acceptance, and impact of the MEMENTO assistive system for dementia patients and their caregivers	1. Assistive devices help with daily functions 2. Setup complexity is a challenge 3. Caregivers provide essential support 4. Personalization enhances use	No measurable cognitive improvement observed
**Holthe et al.** [60]	2022	Systematic review	Norway	Community-dwelling older adults with MCI and dementia	Digital assistive technology	Older community adults with MCI and dementia	Assistive technology	Wearable and assistive technologies to support older adults with MCI and dementia.	To assess advancements in technology use for older adults with mild cognitive impairment	1. Wearables enhance independence by providing reminders for individuals with dementia 2. Apps help with managing daily activities 3. Adoption is rising, but still limited	Supportive role in daily cognition
**Eun et al.** [61]	2022	Quasi-experimental study	South Korea	Older adults with varying cognitive impairments	AI-driven serious game	Older adults with varying cognitive impairments	Wearables (or motion-sensing gaming devices)	AI-personalized therapeutic exercise serious game	To assess the effectiveness of an AI-based personalized serious game in enhancing cognitive and physical abilities	1. AI-based serious games were found to be acceptable, interesting, and motivating for elderly participants. Post-intervention assessments revealed improvements in cognitive function, a reduction in depression levels, and an enhanced quality of life.	Improved cognition and motivation
**Coley et al.** [62]	2022	Randomized controlled trial	Various countries	Online (web-based)	eHealth (web-based platform)	High-, moderate-, and low-engagement groups	ICT platforms/devices	Web-based eHealth intervention with personalized coaching and health tracking features	1. Identifying factors influencing older adults’ engagement with an eHealth intervention 2. Examining its impact on cardiovascular and dementia risk factors	1. Participants who engaged more showed better improvements in cardiovascular and dementia risk factors, including blood pressure, body mass index (BMI), and cholesterol levels 2. Greater interaction (logins, goal setting, coach messages) led to significantly better health	Higher engagement is linked to improvement
**Manca et al.** [63]	2021	Quasi-experimental study	Italy	Controlled laboratory or training setting	Humanoid robots in supporting serious games	Local Train the Brain program.	Robotics/AI-assistive technology	Humanoid robot-based serious games	This study aims to explore the effect of utilizing humanoid robots in supporting serious games for older adults with mild cognitive impairment (MCI)	1. Robot users showed higher levels of engagement and emotional connection 2. Tablet users achieved higher accuracy in correct responses 3. Both groups improved over time, but the robot’s empathic cues appeared to boost motivation	Improved engagement and cognitive training
**Kim et al.** [64]	2021	Randomized controlled trial	South Korea	Community-dwelling or institutionalized	ICT-based Training platform	Community-dwelling or institutionalized	ICT-based system	ICT-based cognitive-physical training	ICT-based training devices, including a virtual reality (VR) bicycle integrated with cognitive training modules such as arithmetic operations, fruit-picking tasks, and puzzle-solving activities	1. Personalized serious games are effective for elderly care 2. Integration of AI enhances therapeutic impact 3. High usability and acceptance among older users 4. Supports dual-target interventions (physical and cognitive)	Improved ADAS–Cog cognitive score (better cognitive performance)
**Kelleher et al.** [65]	2021	Cohort study	USA	MapHabit mHealth app, which assists with cognitive impairments	Tablet-based MapHabit app	MapHabit mHealth app in improving ADL recall	mHealth/eHealth	Assistive technology app featuring personalized visual mapping templates for supporting activities of daily living	To assess the feasibility and preliminary impact of a mHealth assistive technology app in supporting individuals with cognitive impairment in performing ADLs	1. Older adults with cognitive impairments were willing to use the MapHabit mHealth app to assist with activities of daily living (ADLs) 2. After using the app, participants perceived positive effects on functional abilities, social engagement, mood, and memory	Improved memory and ADL performance
**Jung et al.** [66]	2021	Systematic review	South Korea	Home-based or familiar environments	ICT devices (PCs, desk-tops, laptops, handheld devices, and wireless or other devices)	Patients undergoing cognitive assessment	ICT platforms/devices	ICT-based cognitive interventions	To analyze the effectiveness of ICT interventions for older adults with MCI using a systematic review and meta-analysis	1. ICT-based interventions have been shown to significantly improve cognitive function in older adults with MCI 2. The findings support their feasibility, with a call for more rigorous and long-term studies	Significant improvement in cognition
**Diaz-Orueta et al.** [67]	2020	Narrative review	European Union	Not applicable (discussion and review)	Various ICT devices (e.g., robotics, serious games, AR, VR, Smart home systems)	Older adults	Assistive technology	Shaping tech for older adults ethically	To explore the ethical implications, privacy concerns, and autonomy considerations associated with the development and implementation of ICT	1. Ethical considerations are lacking in tech design 2. Older adults prioritize autonomy 3. Privacy and data security concerns	Focus on ethical cognitive support
**Contreras-Somoza et al.** [68]	2020	Cross-sectional study	Eight European countries	Older adults with MCI, informal caregivers, formal caregivers, and administrative staff	EhcoBUTLER ICT platform (tablet-based)	Older adults with MCI, informal caregivers, and formal caregivers	Tablet-based	Feasibility of ICT-based cognitive tools	To assess the acceptability of the EchoBUTLER ICT platform among older adults with MCI and stakeholders involved in their care	1. The EchoBUTLER platform is generally acceptable to older adults with MCI and their caregivers 2. End users find social functionalities particularly valuable	Potential support for cognitive training
**Blok et al.** [69]	2020	Qualitative study	Netherlands	Home and daily life settings	Various ICT devices in everyday use (smartphones, computers, tablets)	Older people with cognitive impairments	ICT platforms/devices	ICT use experience analysis	1. To address how older adults with cognitive impairments use ICT daily 2. To identify the perceived barriers and benefits of ICT usage by older adults	1. ICT use helps enhance social and emotional well-being, facilitating engagement in daily activities 2. Social networks influence ICT use by assisting, initiating, restricting, or enabling the use of shared devices	Supports social and emotional engagement
**Holthe et al.** [70]	2018	Systematic review	Various countries	Home-based and research settings	Various (assistive devices, smart home tech, entertainment, and social tech)	Community-dwelling with MCI and dementia	Wide range/not device-specific	Assistive technology for everyday support	To review technologies explored for older people with MCI/D, usability and acceptability, and involvement of family carers	1. Usability issues impact adoption among older adults 2. User participation enhances the design of technology 3. Standardized assessments are needed for evaluation 4. Accessibility remains a critical concern	Support for aging in place
**Pinto-Bruno et al.** [71]	2017	Systematic review	Various countries	Mixed (community and institutionalized)	ICT-based application in daily living	10 different interventions identified	Smartphone	Digital phenotyping to monitor cognition, mood, and life-space	To assess the validity and efficacy of ICT-based interventions in promoting social health and active aging among people with dementia	1. ICT interventions enhance social participation 2. The lack of standard measures limits the effectiveness of the impact assessment 3. Personalized interventions yield better results	Greater mobility is associated with improved cognitive function
**Malinowsky et al.** [72]	2017	Cross-sectional study	Sweden	Interview-based assessment	Various ET devices for everyday use (unspecified)	Older adults with SCI and MCI	Everyday technologies such as Wearables, computer, phone	Comparing technology use with varying cognitive status.	To investigate and compare self-perceived ability in ET use and the number of ETs used among older adults with SCI, MCI, and controls	1. MCI patients tend to engage less with technology 2. Self-reported usage predicts decline 3. Support programs needed for accessibility	Supports memory and daily functioning

**Table 4 healthcare-13-01421-t004:** ICT adoption enabler–barrier matrix.

Technology	Adoption Enablers (with References)	Adoption Barriers (with References)
ICT Platform	Enhances cognitive function [42,45,46,53] Social interaction [46,49,69] Independence [50,62,66] Support remote care [55,69]	Digital literacy gaps [45,69] Security concerns [46] Mistrust in platforms [53,69]
Tablet	Interactive [43,47] Intuitive interfaces [47,51] Portable and acceptable [65,68]	Training needed [51] User fatigue [64] Customization requirements [68]
Assistive Technology	Supports autonomy [59,60] Daily functioning [60] Personalized care [67]	Ethical concerns [67] Lack of personalization [59] Limited flexibility [60]
Wearable	Continuous monitoring [52,61] Enables early intervention [52,54] Preventive care strategies [72]	Discomfort [61] Accuracy [52] Privacy concerns [72]
Smartphone	Promotes social connectivity [48,71] Easy to use for communication [71] Daily tasks [57]	Interface complexity [57] Cybersecurity risks [48] Lower technology confidence [71]
Telephone	Accessible [44] Emotionally supportive [56]	Limited functionality [44] Not always sufficient for deep support [56]
Robotics/AI Assistive	Therapeutic interaction [63] Enhances emotional and cognitive engagement [63]	Steep learning curve [63] Technology skepticism [63]
mHealth/eHealth	Delivers personalized care [66] Improves health outcomes [66]	Limited awareness [66] Usability challenges [66]
Computer	Home-based training and exercises [58] Promotes engagement [58]	Tech maintenance [58] Complexity for older adults [58]
Other Devices	Monitoring and management [70]	Lack of personalization [70] Insufficient co-design with users [70]

**Table 5 healthcare-13-01421-t005:** Publication bias analysis.

**Publication Bias**		**Coefficient**	**SE**	**95% CI**		**Z**	** *p* **
				**Lower Limit**	**Upper Limit**		
Egger’s regression test	intercept	1.83	0.30	1.16	2.51	6.14	<0.001
	slope	−0.25	0.12	−0.49	−0.01	−2.06	0.04
**Hedge’s**				**95% CI**			
				**Lower Limit**		**Upper Limit**	
Trim and fill	original	0.46		0.35	0.57	-	-
	corrected	0.39		0.28	0.49	-	-

**Table 6 healthcare-13-01421-t006:** Sensitivity analysis of the pooled effect size.

Sensitivity Analysis	Pooled Effect Size (Cohen’s *d*)	I^2^ (%)	Impact on Overall Result
All studies included	0.49	46.0%	--
Excluding high-risk bias studies	0.50	55.4%	Minimal, robust findings

**Table 7 healthcare-13-01421-t007:** Comparison of the current study with existing studies on ICT use in dementia care.

Dimension	Focus on the Current Study	Typical Approaches in Prior Studies
**Study Scope**	Systematic reviews of ICT devices are used to promote cognitive health in dementia care among older people.	Reviews of ICT use in technology for dementia and cognitive treatments.
**Primary Focus**	Investigate the effectiveness of digital engagement and cognitive training in reducing the risk of dementia. Examining differences in access, adoption, and digital literacy. Evaluates user trust and data security in computerized interventions.	It improves cognition, behavior, and social engagement, but lacks long-term evidence to support its effectiveness. Focus on cybersecurity issues in AI-powered cognitive aids. Explore general ICT solutions with multiple devices.
**Key Focuses**	ICT technologies (smartphones, tablets, telemedicine) in dementia care. The effect of computerized interventions on cognitive well-being. Assessment of usability, accessibility, and effectiveness of ICT in supporting individuals with dementia.	The overall impact of technology on healthcare accessibility. Cognitive training devices for the elderly without special reference to dementia.
**Research Gaps**	Limited empirical evidence of the long-term impact of ICT use in dementia and MDI care. Requirement for standardized frameworks to measure cognitive health impacts.	Little attention is paid to their long-term effectiveness in real-world settings. To overcome digital literacy barriers for older adults with cognitive impairment. Lack of integrated ICT systems and multi-device environments.
**Study Contributions**	Performs a systematic review and meta-analysis of the effectiveness of ICT in dementia care. Proposes a concrete framework for evaluating digital interventions for cognitive health enhancement Emphasizes the drivers of ICT adoption among caregivers and individuals with dementia.	Suggests AI-based cognitive health solutions, but they are not validated empirically. The study offers technical guidance for digital health, but it lacks user-focused adoption.
**Data Confidentiality**	Investigates the influence of privacy concerns on ICT uptake in dementia care. Evaluates user trust and data security in computerized interventions.	Considers privacy issues in telemedicine and electronic health records. Focuses on cybersecurity issues in AI-powered cognitive aids.
**User Experience and Accessibility**	Explains how digital interventions may be adapted for dementia patients of varying cognitive abilities. Outlines ICT accessibility barriers among older adults with dementia.	Conducts comprehensive research on the accessibility of technology among aging populations. Fails to incorporate dementia-specific aspects into user-friendly interface design.
**Integration with** **Care Systems**	Explains how ICT solutions may be incorporated into existing healthcare systems for dementia care. Investigates the adoption and training of ICT among healthcare professionals in dementia care.	Explains how ICT solutions may be incorporated into existing healthcare systems for dementia care. Investigates the adoption and training of ICT among healthcare professionals in dementia care.
**Future Directions**	Investigate the need for more research on personalized ICT interventions for each phase of dementia. Research AI-powered cognitive monitoring platforms for improved care. Further research on the long-term use of digital interventions is needed.	Highlights more general digital health interventions that are not explicitly designed for dementia. There is limited evidence of long-term digital intervention use.

**Table 8 healthcare-13-01421-t008:** Summary of limitations and strengths.

Strengths	Limitations
The review followed PRISMA guidelines to ensure a transparent and reproducible methodology.	Many studies used self-reported cognitive data, which may miss subtle impairments and introduce personal bias [47,57].
Covered a broad range of ICT interventions (e.g., tablets, smartphones, mHealth, AI, wearables), showing real-world applicability.	Few studies included long-term follow-up, making it difficult to assess the lasting impact of ICT interventions [43,60].
Focused on diverse user groups, including those with MCI and dementia, to capture varying stages of cognitive decline [42,43,46].	Digital literacy challenges were not thoroughly addressed, particularly among low-income or minority groups [42,46].
Highlighted the need for user-centered and ethically grounded digital solutions, especially in AI-based dementia care technologies.	Ethical issues such as data privacy, transparency, and accountability) In AI-based care, these issues were rarely discussed [58,68].
Emphasized the central role of caregivers in enabling and sustaining ICT use among individuals with cognitive impairment [25,26].	Limited research exists on caregiver support and technological readiness, despite these factors being vital for successful adoption [25,26,59].

## Data Availability

Data are contained within the article.

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
