# Peer review of "Effectiveness, Adoption Determinants, and Implementation Challenges of ICT-Based Cognitive Support for Older Adults with MCI and Dementia: A PRISMA-Compliant Systematic Review and Meta-Analysis (2015–2025)"

_healthcare, 2025, doi:10.3390/healthcare13121421_

Round 1

Reviewer 1 Report

Comments and Suggestions for Authors

The article focuses on mild cognitive impairment (MCI) and dementia, which have become a global health problem with the increasing aging population. In particular, examining the effectiveness of Information and Communication Technologies (ICT)-based interventions for these disease groups represents an approach that is both innovative and has a high social impact.

It has sought a solution to a critical problem in terms of public health, and has made a strategic and timely topic selection in the age of digital health technologies. It has high applicability in developed and developing countries. Since the topic is addressed not only technically but also with cognitive, behavioral, social and ethical dimensions, it is very suitable for health science journals.

In terms of METHODOLOGY, PRISMA (Preferred Reporting Items for Systematic Reviews and Meta-Analyses) compatible systematic review and meta-analysis methodology was used. In addition, a 10-year analysis covering the years 2015–2025 was conducted using a large number of high-quality databases (PubMed, Web of Science, Scopus, IEEE, etc.) with high scientific depth and a careful selection of literature that fully complies with systematic methodology.
The article analyzes 31 eligible studies and classifies 8 types of ICT solutions. The most commonly used technologies are: Tablets, mobile applications, smartphones, wearable devices, robots and AI systems. Lack of digital literacy, hesitations of elderly people regarding use, economic inequalities. Individualized design, support of caregivers, cultural compatibility.
The article analyzes 31 eligible studies and classifies 8 types of ICT solutions.
Dimensions directly related to health such as user experience, accessibility, ethical sensitivities are addressed in a multi-faceted manner, and integration scenarios into health systems are detailed. In addition, current public health issues such as digital exclusion are addressed.

Suggestions

  • The meta-analysis section can be more detailed (such as statistical variances, effect size).
  • A few technical examples/data diagrams regarding the functioning of AI and technology components can be added.
  • A “Limitations and Strengths” table should be provided in a more understandable way.

Author Response

Dr. Lorraine S. Evangelista
Editor-in-Chief
Healthcare
MDPI

Subject: Response to Reviewers – Manuscript ID [healthcare-3657769], “Effectiveness, Adoption Determinants, and Implementation Challenges of ICT-Based Cognitive Support for Older Adults with MCI and Dementia: A PRISMA-Compliant Systematic Review and Meta-Analysis (2015-2025)”

Dear Editor,

Thank you for the opportunity to revise and resubmit our manuscript entitled “Effectiveness, Adoption Determinants, and Implementation Challenges of ICT-Based Cognitive Support for Older Adults with MCI and Dementia: A PRISMA-Compliant Systematic Review and Meta-Analysis (2015-2025)” (Manuscript ID: healthcare-3657769) We greatly appreciate the insightful comments and suggestions provided by you and the reviewers, which have significantly improved the quality of our paper. We have revised the manuscript accordingly and prepared a point-by-point response to each comment. Please find below our detailed responses (in yellow), along with an explanation of the corresponding changes made in the manuscript (highlighted in yellow in the revised version).

We have carefully considered all the suggestions and made significant revisions to enhance the overall quality of the manuscript. We have addressed each comment point-by-point and believe that the revised manuscript has been improved substantially. Once again, we would like to express our sincere thanks for the opportunity and look forward to further possibilities.  

All the queries and responses are attached herewith.

Ashrafe Alam , Md Golam Rabbani, Victor R. Prybutok

Reviewer 2 Report

Comments and Suggestions for Authors

This review examines ICT-based interventions for older adults with MCI and dementia, highlighting benefits in cognition, social engagement, and daily functioning. Drawing on 31 studies published between 2015 and 2025, it categorizes the use of various technologies. The findings highlight that ICT interventions can improve memory, executive function, mood, and daily functioning, particularly when personalized and supported by caregivers. However, there are some points that authors must address.

Major:

  1. Authors should carefully check details, for example, formatting inconsistencies, such as some paragraphs having spaces between them while others do not.
  2. Introduction: here are poor transitions between paragraphs; authors need to improve the flow with smoother transitions. Table 1 is informative but introduced too abruptly, consider moving it to the Discussion section. Research Questions 1 and 3 are overlapping, as both address types of technology and their impact. In R105, the word “nightingales” is unclear and should be rephrased or removed for clarity and professionalism. In R103, the phrase “enable dementia patients to enable caretakers and caregiving” is confusing and should be rephrased for clarity.
  3. Method: Simply stating that “risk of biases was evaluated” is insufficient without specifying the instrument or scoring system used. No methods for meta-analysis are described. Additionally, there is no explanation of which variables were extracted during data collection. Table 3 is not specific to any database.
  4. Analytical Process: Again, there is a lack of meta-analysis.
  5. Discussion: The section lacks a quantitative summary, and the authors overgeneralize claims without sufficient evidence. For example, the phrase “empirical data consistently record substantial improvement...” is not adequately supported.

Minor:

  1. R13–15: The sentence “Systematic reviews and original research articles should have a structured abstract of 13 around 250 words and contain the following headings: Background/Objectives, Methods, Results, 14 and Conclusions” should be removed from the manuscript.
  2. The term “ICT” should be defined at its first appearance in the manuscript.
  3. R24–27: Font size should be consistent throughout the article, and the numbering is out of order.
Comments on the Quality of English Language

The English should be improved for clarity, professionalism, and logical flow.

Author Response

(The authors gave the same response as above.)

Reviewer 3 Report

Comments and Suggestions for Authors

Dear Authors,

Overall Impression

This timely and well-conducted systematic review addresses an increasingly relevant topic: how ICT can aid cognitive abilities in older adults facing dementia or mild cognitive impairment (MCI). The study's adherence to PRISMA standards, alongside its inclusion of meta-analysis, strengthens its methodological basis. That said, the manuscript's impact could be enhanced with some improvements in clarity, a stronger focus on practical implications, and a deeper incorporation of current research.

There are  recommendations for the article “Effectiveness, Adoption Determinants, and Implementation Challenges of ICT-Based Cognitive Support for Older Adults with MCI and Dementia”,

  1. Clarify and Strengthen Study Objectives (Introduction)

The objectives are present but lack precision and differentiation from existing reviews.

Clearly articulate how this review uniquely contributes to the field. Specifically, highlight that this work focuses on ICT-based cognitive support (not general care technologies) and define what types of ICT interventions are included.

  1. Address Long-Term Sustainability of Interventions (Results/Discussion)
    The paper notes that few studies assess the long-term impact but fail to explore the reasons or solutions.
    Include a dedicated sub-section in the Discussion explaining why sustainability data is lacking (e.g., short follow-up, funding limitations) and propose future research strategies to address this, such as longer RCTs or implementation follow-up studies.
  2. Expand Practical Implications for Implementation (Discussion and Conclusion)
    Findings are insightful but not translated into practice-ready strategies.
    Add practical, actionable recommendations for care settings, such as integrating ICT training in caregiver curricula or subsidising devices in community programs. Consider discussing policy-level enablers for broader adoption.

  1. Improve Visual Presentation of Results (Results section)

While the narrative summary is comprehensive, readability would improve with more visuals.

Recommendation:

  • A summary table comparing intervention types and their cognitive effects
  • A bar chart showing effectiveness by intervention category (e.g., tablets, wearables)
  • A matrix showing adoption enablers and barriers across technologies

  1. Contextualize with Post-Pandemic Trends (Introduction and Discussion)

The manuscript covers a 2015–2025 window but barely discusses how COVID-19 changed ICT use among older adults.
Integrate discussion of how the pandemic accelerated telehealth, mHealth, and social tech usage, and cite recent 2022–2023 studies on digital inclusion, remote assessments, or caregiver-mediated digital support.

  1. Include Critical Appraisal Highlights (Results – Quality Assessment)

The JBI risk of bias analysis is mentioned but not illustrated with specific findings.

Add the studies with their JBI scores, and discuss which had the strongest vs weakest methodological rigour and why.

  1. Improve Focus and Avoid Redundancy (Introduction)

The background includes repetitive content on dementia prevalence and technology benefits.
Streamline the introduction by summarizing prevalence and global burden more concisely, and focus on explaining the knowledge gap this review addresses.

Author Response

(The authors gave the same response as above.)

Round 2

Reviewer 2 Report

Comments and Suggestions for Authors

The authors have thoroughly addressed the previous concerns, and the revised manuscript demonstrates improved clarity, methodological rigor, and overall quality. I recommend acceptance.

Author Response

(The authors gave the same response as above.)

Reviewer 3 Report

Comments and Suggestions for Authors

Dear Authors, please find a few more comments below:

Examine the reasons for heterogeneity in more detail. Incorporating a concise meta-regression or a structured subgroup analysis—possibly categorised by the mode of intervention (tablet vs. wearable), study quality levels, or even the extent of dementia—would be advantageous. This exploration would help contextualise why the I² value increased to 55% after removing high-risk studies and ultimately assist in interpreting the overall pooled effect. Quantify your efforts to account for publication bias. While the funnel plot and Egger test (p = 0.04) are compelling, a brief report on a trim-and-fill estimate or a leave-one-out influence plot would offer more profound insight. Readers might then better understand how much the summary effect (d ≈ 0.50) could change when adjusted for any missing "negative" studies. 

When recommending practical steps, link them to relevant cost and equity considerations. The current framework mentions device subsidies and digital literacy training but lacks supporting evidence regarding costs or return on investment. Including one or two economic evaluations (even if just indicative figures) would empower policymakers to assess feasibility.

Specify the precise search end date in the Methods section (e.g., "Searches were last conducted on 31 January 2025") to improve reproducibility. Indicate the statistical software packages used for the meta-analysis (e.g., RevMan, R (meta), Stata). 

Author Response

Dr. Lorraine S. Evangelista
Editor-in-Chief
Healthcare
MDPI

Subject: Response to Reviewers – Manuscript ID [healthcare-3657769], “Effectiveness, Adoption Determinants, and Implementation Challenges of ICT-Based Cognitive Support for Older Adults with MCI and Dementia: A PRISMA-Compliant Systematic Review and Meta-Analysis (2015-2025)”

Dear Editor,

Thank you for the opportunity to revise and resubmit our manuscript entitled “Effectiveness, Adoption Determinants, and Implementation Challenges of ICT-Based Cognitive Support for Older Adults with MCI and Dementia: A PRISMA-Compliant Systematic Review and Meta-Analysis (2015-2025)” (Manuscript ID: healthcare-3657769) We greatly appreciate the insightful comments and suggestions provided by you and the reviewers, which have significantly improved the quality of our paper. We have revised the manuscript accordingly and prepared a point-by-point response to each comment. Please find below our detailed responses (in green), along with an explanation of the corresponding changes made in the manuscript (highlighted in yellow in the revised version).

We have carefully considered all the suggestions and made significant revisions to enhance the overall quality of the manuscript. We have addressed each comment point-by-point and believe that the revised manuscript has been improved substantially. Once again, we would like to express our sincere thanks for the opportunity and look forward to further possibilities.  

All the queries and responses are attached herewith.

Ashrafe Alam , Md Golam Rabbani, Victor R. Prybutok